

# Validity and reliability of a repeated multi-changes of direction agility test in senior soccer players

Mehdi Ben Brahim[1], Adrián García-Valverde[2], Hussain Yasin[1] and Alejandro Sal-de-Rellán[2]

[1] Health and Physical Education Department, Prince Sultan University, Riyadh, Saudi Arabia
[2] Faculty of Health Sciences, Universidad Isabel I de Castillla, Burgos, Castilla y León, Spain

## ABSTRACT

**Background:** This study aimed to evaluate the validity, reliability, and sensitivity of repeated multi-changes of direction agility test (rMCOD) compared to a soccer-specific field test of repeated sprint ability (S-RSA) and repeated sprint ability test (RSA).

**Methods:** Thirty-five healthy male soccer players (age: $18.4 \pm 1.3$ years) from Tunisan national soccer league (elite and sub-elite) took part in this study. They performed the tests in a randomized order over five sessions interspaced by at least 72 h. The construct, predictive and discriminant validity, relative and absolute reliability, and sensitivity of the tests were analyzed. The total and best time of the test (the sum for all trials and the trial with the lowest duration on nine, six, and seven attempts for rMCOD, RSA, and S-RSA, respectively), fatigue index, rating of perceived exertion (RPE), and lactate concentration were recorded.

**Results:** rMCOD correlated with both, S-RSA and RSA in total time ($r = 0.85$ and $r = 0.52$, respectively) and fatigue index ($r = 0.74$ and $r = 0.83$, respectively). Receiver operator characteristics were not able to discriminate between group levels (elite and sub-elite). When comparing training levels, only the fatigue index in S-RSA showed a difference between groups. Fatigue index, total time, and the best time in rMCOD showed *excellent* reliability, as well as the minimal change detectable (MCD = 0.89, MCD = 0.63, and MCD = 0.11, respectively) was higher than the standard error of the mean (SEM = 0.32, SEM = 0.23, and 0.04, respectively).

**Conclusion:** rMCOD showed large to *very large* predictive validity compared with the S-RSA and RSA, being a reliable test for the following parameters: the best time and total time to perform the test. Nevertheless, this study design cannot ensure whether or not this test is able to detect real changes in performance in response to training since it did not include a training intervention; besides, rMCOD could not distinguish between elite and sub-elite players, which is a limitation.

# INTRODUCTION

Soccer is an intermittent team sport characterized by high unpredictability due to its open skill requirements, tactical and physical characteristics (*Gonçalves et al., 2016*).

Corresponding author
Adrián García-Valverde,
adriang.valverde@gmail.com

The physical demands are characterized by a large number of high-intensity actions (*Di Salvo et al., 2007*; *Lago et al., 2010*), which are interspaced with low and moderate-intensity actions over the game (*Stølen et al., 2005*; *Nobari et al., 2023*). However, these demands are influenced by several factors such as the match's intensity (*e.g.*, sprint ability and change of direction) (*Lovell et al., 2018*; *Nobari et al., 2023*); therefore, the performance of these capacities should be known to individualize the training process.

The performance of soccer players is strongly related to change of direction performance and the ability to cover longer distances at very high-speed running (VHSR) (21–24 km·h$^{-1}$), sprinting speed running (>24 km·h$^{-1}$), and the capacity to reach larger numbers of high-speed running (HSR) efforts (>21 km·h$^{-1}$) (*Faude, Koch & Meyer, 2012*; *Rey et al., 2023*; *Varley & Aughey, 2012*; *Rampinini et al., 2007*; *Bloomfield, Polman & O'Donoghue, 2007*). The importance of high intensity actions is based on the fact that most of the decisive actions of the match (*e.g.*, goal) occur in this type of actions (*Faude, Koch & Meyer, 2012*). Per match, a soccer player usually covers a total of 41–568 m at VHSR, 190–236 m at sprinting (*Rey et al., 2023*) and perform around 100–150 accelerations (*Varley & Aughey, 2012*). In this sense, repeated sprint ability (RSA) tests, which are used to assess the ability to perform repeated sprint actions, are also considered a useful test for assessing the players' performance because the results in this test correlate with high-intensity running and sprint distance during soccer matches (*Rampinini et al., 2007*). However, RSA is a test that does not include the wide spectrum of movements required during a match (*e.g.*, jumping, change of direction, running backwards or sideways) (*Bloomfield, Polman & O'Donoghue, 2007*). For this reason, the changes of direction and repeated sprint tests (*e.g.*, the repeated modified agility T-test or 505 change of direction test) were designed to reproduce the specific actions required in soccer, in which players should accelerate and decelerate changing speed and direction during intermittent efforts (*Dugdale et al., 2019*). One of the tests which have been used to assess the changes of direction in soccer player is the Specific Repeated Sprint Ability Test (S-RSA) (*Bangsbo, 1994*). Nevertheless, S-RSA tests (also known as the Bangsbo test) have been set up with nonspecific changes of direction than those performing in the match and do not include other actions (*e.g.*, run backwards or sideways) (*Bloomfield, Polman & O'Donoghue, 2007*). In this sense, a new Multi-Change of Direction Agility test (NMAT) has been proposed as a soccer-specific field test since it allows assessing players' change of direction as it occurs in a match (*Brahim, Bougatfa & Amri, 2013*). Unlike the S-RSA and RSA, the NMAT allows for a specific evaluation of the soccer player since jump, acceleration, braking, and lateral and back displacement are included in the test. However, repeated efforts, which are performed in the game (*Stølen et al., 2005*), are not assessed in its current design, therefore, it seems necessary to explore the possibilities of the NMAT to evaluate repeated sprint actions.

The primary aim of this study is to assess the validity of the test, which represents the ability of the test to actually evaluate what you want to measure (*Hopkins, 2000*); secondly, the inter-day reliability of the test should be assessed with a test-retest procedure (*Atkinson & Nevill, 1998*); finally, the sensitivity of the test, which should show the smallest changes observed to determine changes in performance that the test could detect in an ecological

context (*Currell & Jeukendrup, 2008*). Thus, this study aimed to evaluate the validity, reliability, and sensitivity of the repeated multi-changes of direction agility test (rMCOD) compared to previous validated S-RSA and RSA tests.

## MATERIALS AND METHODS

### Study design

A crossover randomized trial design was applied to assess the validity, reliability, and sensitivity of the rMCOD. Players attended five times on non-consecutive days with at least 72 h of washout between sessions. Each player took a single test during a session. This was started at random times for each participant under the condition that they could not do the same test again in that session. Therefore, in the first session the anthropometric measurement was performed and the players were familiarized with the testing procedure. Sessions two to four were used to test on rMCOD, RSA and S-RSA following a randomization. In the last session, players performed a re-test of the rMCOD.

We recorded the best time, total time (sum of all sprint times), fatigue index (the inverse of the sum of all sprint time divided by the product of the best attempt and number of sprints, also named decrement's score. See Eq. (1)) (*Kerhervé et al., 2020*), rate of perceived exertion (RPE) and peripheral blood lactate concentration so as to assess the validity and reliability of rMCOD (*Impellizzeri et al., 2008*; *Dardouri et al., 2014*).

$$\left| 1 - \frac{\sum_{s=s_1}^{s_n} s}{min\{s_1, \ldots, s_n\} \cdot n} \right| \times 100 \tag{1}$$

Equation (1): Fatigue index equation. Where '$s$' is the time of sprint.

### Participants

Thirty-five male soccer players (elite = 22 from Tunisia's national team- and sub-elite = 13 soccer players who participate in the local championship) (age: $18.4 \pm 1.3$ years; height: $1.8 \pm 0.1$ m; body mass: $76.3 \pm 5.5$ kg; body fat percentage: $11.6 \pm 2.0$%) participated in this study. Body composition measures were taken with a bioimpedance scale (Tanina DC241 MA, IL, USA). Players usually perform five training sessions and an official match per week. No players reported any injury, diseases or intaking of supplements that could influence their performance. All of them signed an informed consent in which the potential risks of this study were explained. The study was conducted according to the Declaration of Helsinki and was approved by the ethics committee of the authors' university before recruitment (University Isabel I of Castile, Ethical Application Ref: UI-PI008).

### Procedures

All players were encouraged to maintain their nutritional routines avoiding caffeine-rich drinks such as coffee or alcohol prior to assessments (in 24 h before testing). Moreover, they were asked to avoid physical exercise before measurement. Every testing session began with a standardized warm-up which included exercises such as jogging, joint mobility, dynamic stretching, and short sprints. All sessions were carried out in April with

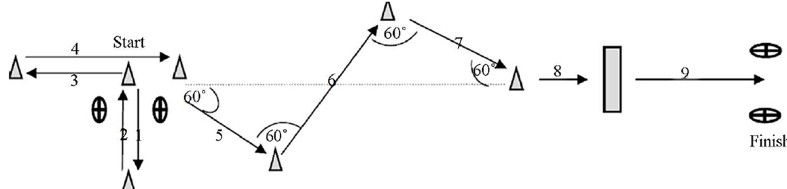

**Figure 1 Schematic representation of the rMCOD.** (1) 2.5 m; (2) 2.5 m; (3) 2.5 m; (4) 3 m; (5) 2 m; (6) 4 m; (7) 2 m; (8) 1.5 m; (9) 5 m.               

similar weather conditions (temperature from 17 to 22 °C) and at the same time (at 8:00 a.m.) to avoid chronobiology bias. Players were verbally encouraged to perform each test as fast as possible.

## Field tests

### Repeated multi-change of the direction agility test (rMCOD)

The rMCOD is based on the new multi-change of direction agility test (NMAT) but performed using repeated sprint actions. The test protocol consists in performing nine repetitions of NMAT with a rest of 25 s between attempts accomplishing a circuit of 225 m in length. Players start with 2.5 m of right-lateral running. Consecutively, they come back doing left-lateral running up to the start place where they need to do 2.5 m of running back, and 3 m forward. Players only needed to cross the marker with one of their feet to consider the try well done. Afterwards, they turn around the marker at 60 degrees to the right to cover 2 m, turns 120 degrees to the left to cover 6 m, and 120 degrees to the right to cover 2 m again. Continuedly, they turn 60 degrees to the left and runs 1.5 m to leap over hurdle at 0.5 m, and sprint 5 m. After each series, players walked back to the start position. Please see the graphic representation of the protocol in Fig. 1. Every attempt was measured by two photocells (Brower timing systems; Draper, UT, USA) placed at the beginning and the end of the circuit.

### Repeated sprint ability (RSA)

Players performed six repetitions of 40 m with a change of direction allowing 20 s of rest between repetitions. The players were required to lead with the front foot 30 cm behind the starting line which was defined by photocell (Brower timing systems; Browner, Draper, UT, USA). Once the players achieved the mark at 20 m from the start line with one of their feet, they had to make a change of direction turning with their preferred leg and coming back to the initial place (*Rampinini et al., 2007*).

### Soccer field test of repeated sprint ability (S-RSA)

The test was based on *Bangsbo (1994)* and modified by *Wragg, Maxwell & Doust (2000)*. Thus, players performed the test by adding a random left or right change of direction over seven repetition with 25 s of recovery between repetition. The players covered 34.2 m per sprint and walked during the recovery to the starting line. Time was measured by photocells (Brower timing systems; Browner, Draper, UT, USA) which were placed at the start line and the end of the track.

## Load assessment

### Rating of perceived exertion (RPE)

Rating of perceived exertion values was obtained using the OMNI-RES scale (*Foster et al., 2001*). This scale aimed to define exercise intensity between "extremely easy" (0) and "extremely hard" (10). Participants were asked, "How hard do you feel the exercise was?" immediately after the last series in each test.

### Peripheral blood lactate concentration

Blood lactate (La) was determined using test strips and a portable analyzer (Arkray Lactate Pro LT-1710–Kyoto, Japan) through peripheral blood samples taken from the earlobe right before the first sprint (baseline) and 3 min after the last series (*Dardouri et al., 2014*). Before extracting the sample, the skin was cleaned with 96° ethanol. For analysis, the two-first blood drops were discarded, and the third drop was used.

## Statistical analysis

Data are presented as mean ± standard deviation (SD). Construct and predictive validity for fatigue index, total time and RPE were assessed through one-way ANOVA and Pearson correlation, respectively. The correlation's coefficient was interpreted according to *Hopkins (2002)* as trivial ($r < 0.1$), small ($0.1 \leq r < 0.3$), moderate ($0.3 \leq r < 0.5$), large ($0.5 \leq r < 0.7$), very large ($0.7 \leq r < 0.9$), and nearly perfect ($r \geq 0.9$). To determine discriminant validity (*i.e.*, elite *vs* sub-elite) a receiver operator characteristics (ROC) curve was used to analyze the area under the curve (AUC) (*Søreide, 2009*). The ROC values were interpreted according to UAC as excellent ($\geq 0.9$), good ($\geq 0.8$), fair ($\geq 0.7$), and non-useful ($<0.7$) (*Carter et al., 2016*). Relative reliability was analyzed using an intraclass correlation coefficient of two-way mixed for absolute agreement based on a single rate ($ICC_{2,1}$) (*Koo & Li, 2016*). $ICC_{2,1}$ was set up with 95% confidence limits (CI). The $ICC_{2,1}$ value was interpreted according to *Portney & Watkins (2002)* as poor ($<0.5$) moderate (0.5–0.74), good (0.75–0.89) and excellent ($\geq 0.90$). Absolute reliability was obtained using an Excel spreadsheet and the equation proposed for standard error of the mean (SEM) (*Hopkins, 2000*; *Atkinson & Nevill, 1998*). To establish the sensitivity of the rMCOD test-retest, it was performed the smallest worthwhile change (SWC) analysis, which was calculated as: $0.2 \times sd(baseline)$ according to the minimal effect size suggested for means difference by *Hopkins (2002)*. Besides, the sensitivity rMCOD test-retest was interpreted by comparing SEM and SWC. At this point, the ability of the test to detect small changes was considered when the SEM was smaller or equal to SWC (*Liow & Hopkins, 2003*). The minimal detectable change (MDC) at the 95% confidence interval was obtained through the formula $\left(1.96 \times SEM \times \sqrt{2}\right)$ (*Beckerman et al., 2001*). RStudio's packages were used for data analysis and the statistical significance was set at $p < 0.05$.

## RESULTS

Relationships between tests are presented in Table 1. The obtained findings for fatigue index revealed positive large and very large relationships between rMCOD performance and S-RSA and RSA, respectively. In addition, confidence interval showed a large to very
**Table 1 Correlation between tests (predictive validity analysis) and 95% confidence interval.**

|  | Fatigue index | RPE | Total time |
|---|---|---|---|
| rMCOD *vs* S-RSA | 0.74 (0.55–0.86)* | −0.06 (−0.38–0.28) | 0.85 (0.72–0.92)* |
| rMCOD *vs* RSA | 0.83 (0.69–0.91)* | −0.06 (−0.38–0.28) | 0.52 (0.22–73)* |

Notes:
rMCOD, repeated multi-change of the direction agility test; S-RSA, specific field test of repeated sprint ability test; RSA, repeated sprint test; RPE, rating of perceived exertion.
* $p < 0.05$.

**Table 2 Comparison between level groups (construct validity analysis).**

|  | rMCOD | | | | S-RSA | | | | RSA | | | |
|---|---|---|---|---|---|---|---|---|---|---|---|---|
|  | Elite | Sub-elite | F | p-value | Elite | Sub-elite | F | p-value | Elite | Sub-elite | F | p-value |
| Fatigue index | 5.15 ± 2.38 | 6.09 ± 3.2 | 0.974 | 0.331 | 5.45 ± 2.37 | 8.07 ± 4.34 | 5.360 | 0.027 | 4.67 ± 2.91 | 6.47 ± 4.28 | 2.202 | 0.147 |
| Total time | 85.4 ± 3.14 | 88.0 ± 4.63 | 3.703 | 0.063 | 59.6 ± 2.04 | 61.0 ± 2.68 | 2.932 | 0.096 | 37.3 ± 1.28 | 37.6 ± 1.31 | 0.491 | 0.488 |
| RPE | 8.5 ± 0.96 | 8.15 ± 1.07 | 0.973 | 0.331 | 8.14 ± 0.83 | 8.54 ± 0.97 | 1.688 | 0.203 | 8.36 ± 1.0 | 8.38 ± 0.87 | 0.004 | 0.950 |

Note:
rMCOD, repeated multi-change of the direction agility test; S-RSA, specific field test of repeated sprint ability test; RSA, repeated sprint test; RPE, rating of perceived exertion.

large relationship in the MCOD-RSA correlation, while in the rMCOD-RSA correlation it showed a large to nearly perfect relationship. Moreover, the total time correlation showed a positive very large and large relationship between rMCOD with both S-RSA and RSA, respectively. Said correlations showed interval confidence from a very large to nearly perfect relationship between rMCOD and S-RSA, and from small to very large in rMCOD and RSA correlation. Only the correlations for fatigue index and total time were significant.

Table 2 shows the comparison between soccer performance groups in each of the tests. The one-way ANOVA only showed significant differences between performance levels in S-RSA for fatigue index. Therefore, said test is able to show differences in fatigue index between elite and sub-elite players (F(1,2) = 5.36; $p = 0.027$).

Figure 2 shows the discriminant validity for all variables in each test. AUC lower than 0.7 were observed for all variables (fatigue index, best time and RPE) and test (rMCOD, S-RSA and RSA).

Table 3 reveals the reliability obtained for the rMCOD. High reliability was observed for the rMCOD. Specifically, ICC for the best time and total time were excellent, above 0.90. The ability of rMCOD to detect small performance changes could be rated as good, given that SWC were higher than SEM for all variables except for delta change of concentration of lactate and rating of perceived exertion.

## DISCUSSION

The main aim of this study was to evaluate the validity, reliability, and sensitivity of the rMCOD compared to S-SRA and RSA tests. This study checked the use the rMCOD as a valid and reliable test to assess the soccer players' repeated sprint ability using sport-specific movements. The main findings in this study indicated that: (i) rMCOD
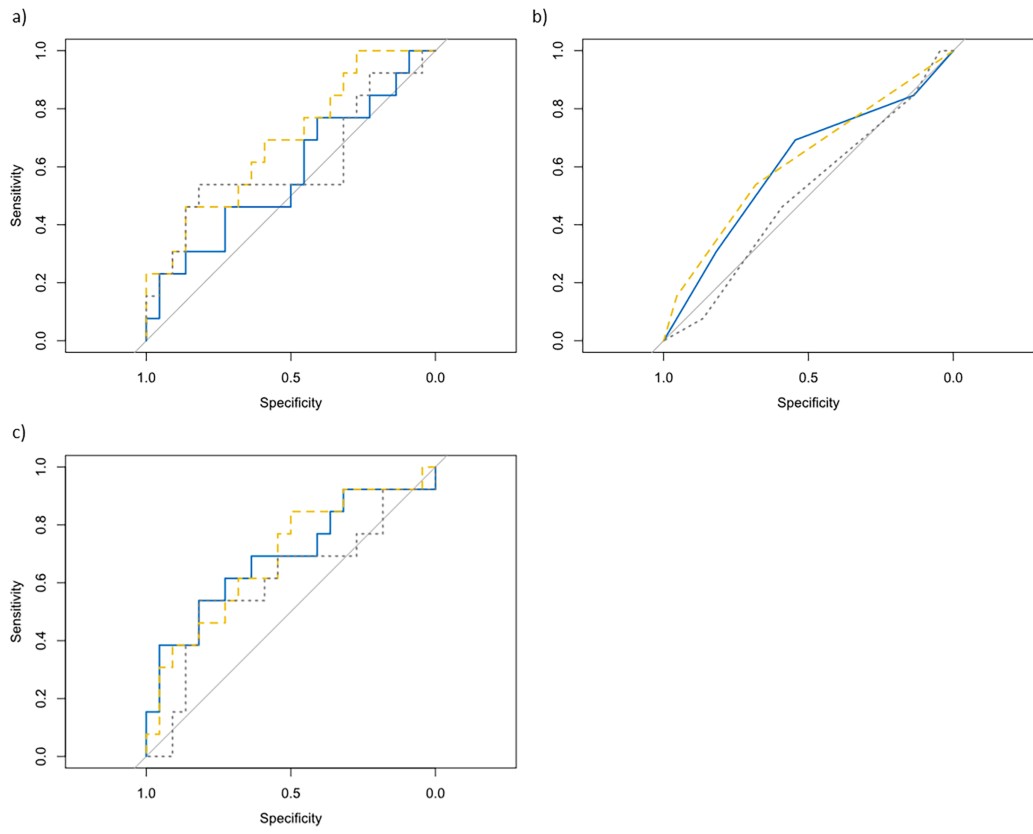

**Figure 2** **Comparison of ROC curve.** rMCOD (blue solid line), S-RSA (yellow dashed line) and RSA (grey dotted line) in (A) fatigue index, (B) RPE and (C) best time (discriminant validity analysis).

**Table 3  Reliability and sensibility analysis results for rMCOD test-retest.**

|  | Trial 1 | Trial 2 | ICC [95% CI] | SEM | SWC | MCD |
|---|---|---|---|---|---|---|
| Fatigue index (a.u.) | 5.50 ± 2.71 | 5.51 ± 2.84 | 0.91 [0.82–0.95] | 0.32 | 0.541 | 0.89 |
| Best time | 9.10 ± 0.43 | 9.13 ± 0.42 | 0.92 [0.84–0.96] | 0.04 | 0.087 | 0.11 |
| Total time | 86.37 ± 3.90 | 86.68 ± 3.66 | 0.96 [0.92–0.98] | 0.23 | 0.78 | 0.63 |
| [La] | 12.86 ± 1.43 | 12.83 ± 1.39 | 0.90 [0.80–0.95] | 0.15 | 0.287 | 0.43 |
| Δ [La] | 6.26 ± 1.65 | 5.93 ± 1.57 | 0.71 [0.50–0.84] | 0.39 | 0.201 | 1.07 |
| RPE | 8.37 ± 1.00 | 8.80 ± 0.87 | 0.05 [−0.25 to 0.36] | 0.34 | 0.331 | 0.95 |

**Note:**
rMCOD, repeated multi-change of the direction agility test; ICC, intraclass correlation; SEM, standard error of the mean; SWC, smallest worthwhile change; MCD, minimal change detectable; [La], concentration of lactate 3-min after testing; Δ [La], increment of concentration of lactate over test; RPE, rating of perceived exertion.

shows reliable test-retest outcomes, (ii) can detect small changes in performance since strong associations were observed between rMCOD with S-RSA and RSA test in fatigue index and total time, but it is not able to differentiate between elite and sub-elite players' performance level.

The development of soccer players' physical fitness is one of the main aims of strength and conditioning coaches in this sport. However, the assessment of its components is a

complex task which should be evaluated through several tests like S-RSA and repeated shuttle-sprint test (*Altmann et al., 2019*). Given the importance of the repeated change of direction ability in soccer, it is important to apply specific tests which really measured this skill in a player's assessment. Currently, S-RSA (*Bangsbo, 1994*) and NMAT (*Brahim, Bougatfa & Amri, 2013*) are commonly used to assess the change of direction in soccer players. However, NMAT only assesses one sprint, whilst players need to do it multiple times during the match (*Chaouachi et al., 2012*). That is why rMCOD, based on NMAT, has been proposed as an integral test of change of direction. Indeed, the significant correlation between rMCOD and S-RSA could support the predictive validity (*e.g.*, the performance in the test is correlated with a criterion measure) of the rMCOD. While previous tests have been validated for evaluating isolated capacities such as change of direction or RSA in soccer players, these tests are not able to assess both capabilities together in a sport-specific environment.

Likewise, rMCOD showed stability of a measure under repeated measurement, therefore, elite and sub-elite soccer players could be precise enough to observe changes in performance (*Noble, Scheinost & Constable, 2021*). Specifically, time measurements (best time and total time) have shown high reliability scores. In addition, the relationship between SEM and SWC demonstrated the usefulness of rMCOD. The SEM values were lower than SWC, therefore, rMCOD can detect small changes in performance. Then, time measurements, fatigue index, and lactate concentration could be considered reliable. Nevertheless, this research does not ensure that the rMCOD is capable of identifying changes in performance due to training since only acute responses were assessed.

The MDC values provided in this study indicated that changes in rMCOD performance beyond them could be considered a real change, although differences were not found between players' levels. Thus, the discriminant validity as the ROC curve confirmed showing AUC < 0.70, suggest that rMCOD cannot be used to sort between high or low-performance players. Therefore, rMCOD should be validated with a higher number of players of different levels including professional and non-soccer players (*Thomas et al., 2022*). Since the level performance is close between elite and sub-elite, they could show a similar level of performance in running tests (*Ooi et al., 2009*; *Obetko et al., 2019*).

On the other hand, the results of this study should be cautiously interpreted due to limitations. rMCOD should be compared with a specific skill test (*e.g.*, jump test, change of direction or sprint) to ensure that rMCOD is related to those abilities. For this reason, the strength and biomechanical test should be performed in the future to achieve further information about the association with rMCOD. In addition, the shown correlation is not synonymous with causation. Therefore, the association between S-RSA and rMCOD simply shows the magnitude of interrelation between these tests. Finally, this study should have had a larger number of participants from various categories, which could have ensured the validity of the test.

## CONCLUSION

The rMCOD reported high reliability and predictive validity in soccer players compared with the S-RSA and RSA, being a reliable test for the following parameters (*i.e.*, the best

time and total time to perform the test). Moreover, the rMCOD showed that it was sensitive to small differences in performance, however, this test cannot distinguish between elite and sub-elite players, which is a limitation to consider when using this test. Our results suggest that the rMCOD could be an interesting option in the field of sports performance although should delve into its accuracy.

### Funding
The authors received no funding for this work.

### Competing Interests
The authors declare that they have no competing interests.

### Author Contributions
- Mehdi Ben Brahim conceived and designed the experiments, performed the experiments, authored or reviewed drafts of the article, and approved the final draft.
- Adrián García-Valverde analyzed the data, prepared figures and/or tables, authored or reviewed drafts of the article, and approved the final draft.
- Hussain Yasin conceived and designed the experiments, performed the experiments, authored or reviewed drafts of the article, and approved the final draft.
- Alejandro Sal-de-Rellán analyzed the data, prepared figures and/or tables, authored or reviewed drafts of the article, and approved the final draft.

### Ethics
The following information was supplied relating to ethical approvals (*i.e.*, approving body and any reference numbers):

The University Isabel I of Castile granted Ethical approval to carry out the study within its facilities (Ethical Application Ref: UI-PI008).

### Data Availability
The raw data is available in the Supplemental File.

### Supplemental Information
Supplemental information for this article can be found online at http://dx.doi.org/10.7717/peerj.16753#supplemental-information.

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
