# Peer review of "Validity and reliability of a repeated multi-changes of direction agility test in senior soccer players"

_PeerJ, doi:10.7717/peerj.16753_

## Round 0.1 · original submission · Major Revisions

Both reviewers found merit in your manuscript but have provided feedback to improve the paper. There are several places where reviewers recommended rewriting the sentence for clarity.

**Language Note:** The review process has identified that the English language must be improved. PeerJ can provide language editing services - please contact us at copyediting@peerj.com for pricing (be sure to provide your manuscript number and title). Alternatively, you should make your own arrangements to improve the language quality and provide details in your response letter. – PeerJ Staff

Reviewer 1 ·

Basic reporting

Abstract
Line 27-28: Please revise the “index fatigue” to “fatigue index”.
Line 29: “training level comparison” might be a better term to use.
Line 35-36: Please revise/clarify the following sentence “…and it can detach real changes in performance”.
Introduction
Line 44-45: Please provide examples of “playing levels”. What do you mean by playing levels or players’ levels?
Line 44-45: Please clarify what you mean by “the assessment of these capacities should be performed to sort the players’ levels”.
Line 46-47: Instead of addressing different field tests (which is irrelevant to the topic), I would suggest to focus on repeated sprint ability (e.g., address why the ability to repeat sprint is crucial in soccer players, then discuss previous tests to measure repeated sprint ability).
Line 50: Please clarify what you mean by “…considered valid…”. Does it mean that RSA is considered valid as an indicator of match-related physical performance?
Line 52: Providing examples of “wide spectrum of movements required during a match (e.g., ……..)” might help readers to understand the limitation of RSA.
Line 53-56: Sassi et al. (reference #9) does not discuss repeated modified agility T-test and 505 change of direction test. Please revise the reference.
Line 58: Please clarify “larger turns”.
Line 59: Providing examples of “other actions” might be helpful.

Experimental design

Methods
Line 83: Please explain further about “total time”. Is it the sum of 9 sprints time?
Line 83: Please elaborate on “fatigue index”. What do you mean by “as a percentage of performance deceleration in between sprints”?
Line 88: Please revise the following – “height: 1.8 ± 0.1 cm”.
Line 88: Please provide detail on how the body fat percentage was examined.
Line 95-99: Were there any requirements (or restrictions) before each visit, such as no exercise & alcohol 24h, no caffeine 4 or 8h, and no food consumption 2 hours prior to each visit (to standardize between visits). If there were pre-visit requirements, please provide them in the manuscript. If not, do you think this can be a potential limitation for your study?
Line 101-109: Please provide the following information: what did participant do during 25 seconds of rest between sprints (jogging back to the starting position? Or walking back?)
Line 101-109: At each markers (where participants had to change directions), did participants had to touch the base of markers before moving on or turn around the markers?
Line 101-109: Please provide how the time was measured.
Line 103-104: It might be useful to explain why “9 repetitions of NMAT with a rest period of 25 seconds between sprints” was chosen for the present study.
Line 111-116: Please provide more detail on RSA. For example, did the participants have to touch the marker at 20 m or turn around the marker? Were they allowed to make a turn with their preferred leg or did they have to alternate their leg for making turns between sprints (e.g., right then left).
Line 117-121: Please provide more detail on S-RSA. For example, what was the distance for each sprint? What was the protocol during 25 seconds of recovery? How was the randomization of direction decided?
Line 128-133: Please clarify when the blood lactate was taken for the basal. It was mentioned that it was before the first series. Does it mean that it was taken right before the first sprint (after warming-up) or before the warming-up?
Line 140-141: Please revise UAC to AUC (area under the curve).
Line 147-148: Please provide how SWC was calculated for the analysis.

Validity of the findings

Results
Line 155-172: Please elaborate the results from table 1, 2, 3 and figure 2. Although all results are presented in tables and figures, some readers might find it difficult to interpret the results from the them (e.g., p-value, 95% CI, etc.).
Line 161-162: Once again, please elaborate the results. Did one-way ANOVA showed significant differences in all variables between performance levels in S-RSA? What was the variable that showed significant difference in S-RSA.
Line 161-162: Mentioning (i.e., elite vs. sub-elite) might be helpful for readers to understand what between-group comparison indicates.
Discussion
Line 179-181: Please clarify the main finding ii. i.e., “can detect small change performance the rMCOD have a strong association with S-RSA and RSA test,…”.
Line 186-188: Please revise the following sentence “However, they only assess one sprint”. S-RSA includes repeated sprints while NMAT only assess one sprint.
Line 190-191: I recommend to add a sentence explaining what the result (from concurrent validity) indicate (what does it mean?).
Line 199-201: The following sentence appears to be overstated: “…can be confident that changes in performance are due to athlete’s improvement”. Because the test is considered reliable, does it really mean that the changes in performance are due to athlete’s improvement?
Line 207: Please revise the following part: “professional and non-player soccer”.
Line 207-208: What was the criteria (difference) used to categorize participants into elite and sub-elite? This might be helpful for readers if provided here or in method section.
Line 174-216: Some variables (e.g., RPE and lactate) are measured in the present study but never addressed in the discussion. Please also discuss findings on those variables in the discussion section.
Conclusion
Line 223-224: The following sentence is not related to the findings of the present study: “…rMCOD could be a relevant option not only in the field of sports performance, but also in rehab programs”.

Reviewer 2 ·

Basic reporting

Line 20: “soccer-specific”
Line 23: “interspaced”
Line 24: “At least 72 hours” or “72 hours”; “of resting” needs to be replaced or removed because it leads reader to believe participants were not allowed to engage in daily occupational activities such as walking to work, etc.
Line 25: First sentence of the Results section in the abstract seems to fit better in the Methods section of the abstract.
Line 27: “index fatigue” needs to be replaced with “fatigue index”
Line 29: “on level training comparison” needs to be replaced with some other phrase like “When comparing training levels,”
Line 31: “best time” could be better explained previously in the abstract. Is the best time the trial with the lowest duration? How many trials were performed? The lack of information makes it hard to comprehend the abstract.
Line 35: remove the parenthesis in this sentence and use a “following parameters: the best time and total time to perform the test”
Line 35: “detach” needs to be replaced with “detect”
Line 41: rewrite first sentence to avoid saying “and” multiple times.
Line 44: “playing level” seems to refer to the level of competition and, besides in the research setting, the assessment of physical capacities is to allow for tracking process and evaluating the need for interventions. Comparing players from different levels of competition is typically not the goal in applied settings. I recommend rewriting the last sentence in the first paragraph to indicate why testing physical performance is important in sports.
Line 58: “these tests have been set up with larger turns than performing in the match and not include other actions” needs to be rewritten for clarity.
Line 61: “Opposite S-RSA and RSA,” needs to be replaced with another phrase. Think about something like “Unlike the…” etc.
Line 66: Rewrite the first sentence. I would recommend something like “The primary aim of this study is to assess the validity of the test….”
Line 77: replace “non-following” with “non-consecutive”
Line 78: replace “non-respawn” with another term.
Line 80: remove “from” in the sentence. “Sessions two to four were…”
Line 82: Rewrite this sentence for clarity “To assess the validity and reliability of rMCOD, the best attempts as a time measurement and the total time, the fatigue index was recorded as a percentage of performance deceleration in between sprints [17], as well as the rating of perceived exertion (RPE) and peripheral blood lactate concentration.”
Line 87: “from the Tunisian national soccer league”
Line 88: height seems to be in meters instead of centimeters as indicated.
Line 91: “All of them wrote an informative consent in which the potential risks of this study were explained.” Sentence needs to be rewritten. Did the athletes sign informed consent forms or write them?
Line 96: “exercise as jogging” could be replaced with “exercises such as jogging, etc.
Line 106: “where initials” needs to be rephrased.
Line 113: Use past tense. “The players were required to lead with the front foot 30 cm”
Line 118: Extra space in the beginning of the sentence.
Line 118: Rephrase “In this sense”
Line 120: “(Hidrofit®, Belo Horizonte-MG, Brazil)” the font in this text is not black, it seems like grey.
Line 131: “baseline” instead of “basal”.
Line 145: “obtained using a spreadsheet” Does this refer to an Excel spreadsheet? Please include which program was used.
Line 152: “was set up at” needs to be replaced by “was set at”
Line 161: Rewrite the first sentence “Table 2 are presented the between-group comparison regarding physical tests.”
Line 164: “An AUC lower than 0.7 were observed for all variables (fatigue index, best time and RPE) and test (rMCOD, S-RSA and RSA).” needs to be revised for grammatical issues. “AUC values lower than 0.7 were observed…”
Line 172/ Table 3: Please readjust the table’s margins so that the numbers and words fit within the same line. It is okay if the SD’s are one line below, but at least on my version, there are some decimals on the line below.
Line 176: Remove “tries to provide evidence”
Line 179: “Small changes in performance”
Line 188: “make it repetitively” should be rewritten.
Line 193: “sport-specific environment”
Line 196: “have shown high reliability scores”
Line 197: “the relation between SEM and SWC showed the usefulness of the rMCOD” should be rewritten for clarity.
Line 204: Replace “In this sense”
Line 2012: “in the future”
Line 214: “these tests”
Line 207: “non-player soccer” should be replaced with “non-soccer player”

Experimental design

Line 25: what does “total and best time of the test” mean?
Lines 117-121: Please explain in further detail the S-RSA test.
Line 121: Why were the timing gates different for the RSA and S-RSA tests?
Line 133: Which timepoints was lactate collected?
Line 136: Please provide the variables which were used to assess construct and concurrent validity in the one-way ANOVA and Pearson correlations.
Line 143: You may need to double check on this but I believe the ICC of two-way mixed-effects model for absolute agreement should be used (Koo & Li, 2016).
Line 160/ Table 1: Concurrent validity implies that the measures were assessed at the same time. I believe the type of validity assessed when correlating the performance across tests would be predictive validity.
Line 163/ Table 2: Table 2 does not provide much information besides the fact that the two playing levels did not perform differently. However, it does not provide the means and standard deviations for the groups. I suggest completely rearranging this table.
Line 166/ Figure 2: There are no figure legends or labels. The figure also did not include a caption. It is unclear which variables were represented in each of the three figures.
Line 170: I believe calling the variable “increment of concentration of lactate” may be misleading to a reader. A term like “delta change of lactate concentration” or “post-pre change in lactate” would probably work better.

Validity of the findings

Line 28: why are there two r values for the relationship between rMCOD and fatigue index?
Line 36: Please provide insight into why the authors claim that the rMCOD can detect “real changes” in performance when athletes were not trained over time for potential changes in performance to occur. Further, rMCOD could not distinguish between elite and sub-elite players.
Line 200: Based on this study design, one could not confidently affirm that if a training study was conducted using rMCOD, the “changes in performance are due to athlete’s improvement.”
Line 224: “but also in rehab programs.” There was no mention of this test being used in rehabilitation settings in the discussion, however, that is the last sentence of the conclusion.

---

## Round 0.2 · Major Revisions

There are still come points that need to be made clearer for both reviewers. Some of these are grammatical but one was related to greater clarification about what defines elite vs. sub-elite.

Reviewer 1 ·

Basic reporting

no comment

Experimental design

no comment

Validity of the findings

no comment

Additional comments

The authors have addressed all of my concerns/comments with the original manuscript.
I have a few suggestions that the authors might want to address further.

Abstract
- I think the following sentence in the abstract can make readers confusing: "Nevertheless, this study design cannot ensure that this test might be able to detect real changes in performance since it was not done a large training time which provide these evidence". It might be better to delete this sentence and provide a sentence to interpret the main finding (i.e., what does the main finding indicate/mean?).

Introduction
- The last sentence of the first paragraph is still confusing: "However, these demands are influenced by several factors such as level of competition during the match (e.g. sprint ability, change of direction,) [6,7], therefore, the performance of these capacities should be known to individualize the training process". I recommend to revise/clarify this sentence. (E.g., what do you mean by the level of competition during the match?).

Method
- The authors addressed the restrictions prior to the visit (e.g., stimulant drink, exercise, etc). It might be better to provide the time for those restrictions (e.g.. refrain from exercise for how many hours before the visit, was it only for the testing day?)

- For S-RSA, how many repetitions were performed?

Reviewer 2 ·

Basic reporting

Line 22 - Tunisian
Line 24 - "interspaced by at least 72 hours"; during my last review, I don't think I made it clear that I meant to place the "by" before the at least. I apologize about that.
Line 30 - "Receiver operator characteristics WERE not able"
Line 38 - I believe the last sentence of the paragraph could be improved. Here's a suggestion (you can write it your own way, this is merely a suggestion): "Nevertheless, this study design cannot ensure whether or not this test is able to detect real changes in performance in response to training since it did not include a training intervention; besides, rMCOD could not distinguish between elite and sub-elite players, which is a limitation."
Line 47 – “(e.g. sprint ability, change of direction,)” should be replaced with “(e.g., sprint ability and change of direction)”
Line 50 - “The performance of soccer players is strongly related to change of direction and the ability to cover a great distance to very high speed running (VHSR) (21-24km·h-1), sprinting speed running (>24km·h-1) and the capacity for to reach a great number of high speed running (HSR) efforts (>21km·h-1),” should be rewritten. My suggestion: “The performance of soccer players is strongly related to change of direction performance and the ability to cover longer distances at very high-speed running (VHSR) (21-24km·h-1), sprinting speed running (>24km·h-1), and the capacity to reach larger numbers of high-speed running (HSR) efforts (>21km·h-1)”
Line 57 – “a useful” instead of “an useful”
Line 60 – “e.g.,” add commas to e.g. throughout the paper
Line 61 – “sideways” and “For this reason,”
Line 68 – “e.g.,” add commas to e.g. throughout the paper ; “sideways”
Line 75 – the font type and does not match the rest of the paper. It switches from Times New Roman to Verdana
Line 89 – there are two periods after the word “session”
Line 89 – “it was carried out an anthropometric analysis” should be rewritten
Line 92-93 – “sum of all sprint times”
Line 93 – “(the inverse of sum of all sprint time divided by the product of the best attempt and number of sprints, also named decrement score)” this is merely a suggestion, but I am wondering whether it would be easier to just add the formula here instead of writing it.
Line 125 – “run 1.5 m”
Line 132 – font switches to Verdana again
Line 166 – “an Excel”
Line 183 – “confidence interval”
Line 187 – “comparison between the elite and sub-elite groups in each of the tests.” (there are two periods at the end of the sentence and the term ‘level groups’ sounds confusing)
Line 190 – provide the actual p-value rather than p > 0.05
Line 194/ Figure 2 – What do the different colored lines represent in the figure? Figure 2 presents some of the main results. Efforts should be made to make this figure easier to comprehend.
Line 205 – “sport-specific”
Line 207: “small changes in performance”; “since strong associations were observed between rMCOD with the S-RSA and RSA tests in fatigue index and total time”
Line 216 – “multiple times” rather than “all the time”
Line 219 – “e.g.,” add commas to e.g. throughout the paper
Line 228 – replace “in this sense” with another term
Line 229 – “reliable, Nevertheless,” should be edited or turned into two separate sentences.
Line 230 – “this research does not ensure that the rMCOD is capable of identifying changes in performance due to training since only acute responses were assessed.”
Line 232 – replace or rewrite the sentence that says “non-difference”
Line 233 – “discriminant validity,”
Line 234 – “players”
Line 236 – rewrite the part that says “since closer level performance”
Line 240 – “For this reason, strength and biomechanical tests should be…”
Line 244 – “tests”
Line 245 – rewrite this sentence for clarity
Line 250 – “small differences in performance” instead of “changes”
Line 252 – “when using this test” instead of “when this be used”
Line 253 – rewrite the last sentence for clarity

Experimental design

Line 23 and 98 - you provide that the difference between elite and sub-elite is that one group had signed with a professional club and the other did not. However, you mentioned that all players are from the Tunisian national soccer league. How does that work? If they haven't signed with a professional team, then how can they be part of the Tunisian national soccer league? If they are not part of the Tunisian soccer league, then where do they play/ practice? What is the level of competition of these sub-elite players?

Validity of the findings

None

---

## Round 0.3 · accepted · Accept

The authors have addressed the reviewers' concerns. The manuscript is ready for publication.

Reviewer 2 ·

Basic reporting

N/a

Experimental design

N/a

Validity of the findings

N/a